# Altered topology of neural circuits in congenital prosopagnosia

**Gideon Rosenthal[1,2]\*, Michal Tanzer[2,3], Erez Simony[4,5], Uri Hasson[6], Marlene Behrmann[7], Galia Avidan[1,2,3]\***

[1]Department of Cognitive and Brain Sciences, Ben-Gurion University of the Negev, Beer-Sheva, Israel; [2]The Zlotowski Center for Neuroscience, Ben-Gurion University of the Negev, Beer-Sheva, Israel; [3]Department of Psychology, Ben-Gurion University of the Negev, Beer-Sheva, Israel; [4]Faculty of Electrical Engineering, Holon Institute of Technology, Holon, Israel; [5]Department of Neurobiology, Weizmann Institute of Science, Rehovot, Israel; [6]Department of Psychology and the Neuroscience Institute, Princeton University, Princeton, United States; [7]Department of Psychology and Center for the Neural Basis of Cognition, Carnegie Mellon University, Pittsburgh, United States

**Abstract** Using a novel, fMRI-based inter-subject functional correlation (ISFC) approach, which isolates stimulus-locked inter-regional correlation patterns, we compared the cortical topology of the neural circuit for face processing in participants with an impairment in face recognition, congenital prosopagnosia (CP), and matched controls. Whereas the anterior temporal lobe served as the major network hub for face processing in controls, this was not the case for the CPs. Instead, this group evinced hyper-connectivity in posterior regions of the visual cortex, mostly associated with the lateral occipital and the inferior temporal cortices. Moreover, the extent of this hyper-connectivity was correlated with the face recognition deficit. These results offer new insights into the perturbed cortical topology in CP, which may serve as the underlying neural basis of the behavioral deficits typical of this disorder. The approach adopted here has the potential to uncover altered topologies in other neurodevelopmental disorders, as well.

**\*For correspondence:** gidonro@gmail.com (GR); galiaa@bgu.ac.il (GA)

**Competing interests:** The authors declare that no competing interests exist.

## Introduction

Understanding the neural basis of developmental disorders such as congenital prosopagnosia (CP) remains a challenge from both a basic science and a translational perspective as there are no obvious identifiable deficits on conventional anatomical MR brain scans. Furthermore, many studies show that CP individuals evince normal fMRI activation in the 'core' face-related posterior patches of the brain (*Hasson et al., 2003*; *Avidan et al., 2005*, *2014*; *Avidan and Behrmann, 2009*) (but see *von Kriegstein et al., 2008*; *Dinkelacker et al., 2011*; *Furl et al., 2011*). In contrast, more sensitive methods that have been used to map structural changes in CP relative to controls, such as diffusion tensor imaging (DTI) have revealed a reduction in long-range white matter tracts connecting the 'core' face-related posterior patches and the anterior temporal lobe face patch (ATL) in CP (*Thomas et al., 2009*; for related papers, see *Grossi et al., 2014*; *Scherf et al., 2014*). Other studies have also reported local structural and functional atypicalities in the vicinity of face-selective regions (*Gomez et al., 2015*; *Song et al., 2015*; *Lohse et al., 2016*). Using standard functional connectivity (FC) analysis, which measures the temporal correlations across different brain areas within an individual, we have previously documented abnormal deviations from the control pattern in the connectivity patterns between the 'core' and 'extended' nodes of the face system (*Avidan and Behrmann, 2014*).

**eLife digest** Human babies prefer to look at faces and pictures of faces over any other object or pattern. A recent study found that even fetuses in the womb will turn their heads towards dots of light shone through the mother's skin if the dots broadly resemble a face. Brain imaging studies show that face recognition depends on the coordinated activity of multiple brain regions. A core set of areas towards the back of the brain processes the visual features of faces, while regions elsewhere process more variable features such as emotional expressions.

Around 2% of people are born with difficulties in recognizing faces, a condition known as congenital prosopagnosia. These individuals have no obvious anatomical abnormalities in the brain, and brain scans reveal normal activity in core regions of the face processing network. So why do these people have difficulty with face recognition?

One possibility is that the condition reflects differences in the number of connections (or "connectivity") between brain regions within the face processing network. To test this idea, Rosenthal et al. compared connectivity in individuals with congenital prosopagnosia with that in healthy volunteers. In the healthy volunteers, an area of the network called the anterior temporal cortex was highly connected to many other face processing regions: that is, it acted as a face processing hub. In individuals with congenital prosopagnosia, this hub-like connectivity was missing. Instead, a number of core regions involved in processing the basic visual features of faces, were more highly connected to one another. The greater this "hyperconnectivity", the better the individual's face processing abilities.

The findings of Rosenthal et al. pave the way for developing imaging-based tools to diagnose congenital prosopagnosia. The same approach could then be used to investigate the basis of other neurodevelopmental disorders that are thought to involve abnormal communication within brain networks, such as developmental dyslexia.

The pattern of FC within each individual, as utilized in previous studies, is a combination of stimulus-induced correlations, intrinsic neural fluctuations, and correlations induced by non-neuronal artifacts (such as head motion, respiration). Separating these factors is challenging within the framework of standard FC, given the strength of the intrinsic neural fluctuations. Hence, group differences in FC may not be sufficiently robust to be detected following whole brain statistical correction, and, therefore, are less suitable for mapping large-scale changes in network topology.

In contrast with these previous studies that have examined the neural profile of CP based on a subset of brain regions and their connectivity, here, we adopt an innovative, large-scale network approach. The primary goal of this approach is to elucidate the functional brain topology in individuals with CP (and matched controls) as a means of examining alterations in neural circuitry. Because there is consensus that multiple regions (face 'patches') are implicated in normal face recognition in humans (*Haxby et al., 2000*; *Pyles et al., 2013*; *Weiner and Grill-Spector, 2013*) and in non-human primates (*Tsao et al., 2006*; *Hung et al., 2015*; *Moeller et al., 2017*), elucidating alterations in the topology of this distributed cortical circuit is of great interest.

To that end, in the present study, we have used a novel method, termed 'inter-subject functional correlation' (ISFC), which is designed to isolate stimulus-locked functional responses, by correlating the response profile across the brains of multiple participants (*Simony et al., 2016*). Importantly, intrinsic neural dynamics during rest and during task conditions that are not related to the pattern of activation evoked by stimulation, as well as non-neuronal artifacts (e.g., respiratory rate, motion), can only influence the pattern of correlations within each individual brain, but cannot induce correlations between subjects. In contrast, neural processes that are locked to the structure of the stimulus can be correlated across brains. Thus, the ISFC method allows us to track stimulus-locked brain responses within the high-level visual network in control subjects and to contrast these correlation patterns with the patterns uncovered in CP individuals. Such an approach enables us to explore possible alterations in connectivity across large swaths of the cortex in an assumption-free manner rather than focusing on a predetermined subset of regions and connections.

## Results

During an fMRI scan, 10 CP and 10 control subjects viewed separate blocks of images of emotional faces (angry, fearful), neutral faces, famous faces, unfamiliar faces and buildings (*Avidan et al., 2014*). To define an initial set of unbiased nodes (functional regions or clusters comprising the network) with which to explore topology and connectivity, face-selective (right FFA) and non-face selective (right LOC) seed regions were defined based on BOLD data from a separate group of 16 control subjects (see Materials and methods). Using these two seed regions, and their mean activation profile across subjects, two correlation maps were derived and a binary mask was constructed for each. These masks were then separately sub-divided into small, spatially constrained clusters with each cluster serving as a node for the network analyses. This procedure resulted in large swaths of cortex sub-divided into nodes, each of which preserved the original 'functional tagging' from the seed correlation analysis (i.e., face-selective nodes, orange color, object-selective nodes, green color, or nodes that are not exclusive to either faces or objects, blue color; see Materials and methods for details on node definition, see *Figure 1*).

### Inter-subject functional correlation (ISFC) analysis

To detect stimulus-locked changes in functional responses to faces in CP relative to controls, we used the inter-subject functional correlation (ISFC; see Materials and methods), which calculates the inter-regional correlations in the brains of different individuals who viewed the same stimuli (*Simony et al., 2016*). *Simony et al. (2016)* demonstrated that the ISFC method substantially increases the signal-to-noise (SNR) ratio in detecting shared stimulus-locked network correlation patterns by filtering out idiosyncratic intrinsic neural dynamics and non-neuronal artifacts (e.g., respiratory rate; motion) that can influence FC patterns within a brain but that are uncorrelated across the brains of different participants. Capitalizing on the high SNR of the ISFC procedure permits the construction of a fine-grained functional brain network even with a relatively small sample size as is the case in the present study and potentially in other situations of relatively rare disorders (see Materials and methods).

We calculated the ISFC within the CP and the control groups using the BOLD signal elicited in response to faces and buildings (see below). For comparison, we also ran the same analysis using a standard FC procedure. In the FC analysis, the response profile in each node was correlated with the response profile of all other nodes within an individual. The analysis was repeated for each individual in each group and statistical significance for each edge was determined using a t-test followed by FDR correction. We return to these results below. ISFC is similar in logic to FC, with one critical difference: instead of correlating the response profile within the brain of each individual, we calculated the correlation patterns across brains. For each experimental group (controls and CPs), the correlation between the response profile of each subject and the mean of the remaining 9 subjects was calculated (see ISFC procedure in Materials and methods). The average non-thresholded networks for each group are presented in *Figure 1a,b* for visualization purposes.

As is evident, the raw mean networks over the ISFC of each group are visually similar. To evaluate whether there are any statistical differences in the network structure across the two groups, the networks were directly compared using a permutation test (see Materials and methods for details). This analysis resulted in a difference network in which the edges indicate the significant difference between the two groups (either controls>CPs or CPs>controls). Non-significant edges were eliminated (see details of statistical analysis in Materials and methods ISFC Formulation section). For the following analysis, we applied a stringent statistical criteria. This approach is followed by a second, more descriptive network analysis which captures the overall difference in the pattern of ISFC across the two groups (see Network analysis section).

The controls>CPs difference network revealed that control participants, but not CP, exhibited an overall, non-selective increase in ISFC patterns from nodes in the vicinity of the ATL (see *Table 2* and Network analysis section). In contrast, the opposite analysis of CPs>controls revealed a significant edge between nodes in the vicinity of the LOC (see *Figure 1c,d*).

### Correlations of the significant edges with behavior

To examine whether the neuronal ISFC of these significant edges is related to behavioral performance, the mean raw ISFC value for each group was correlated with the famous faces questionnaire

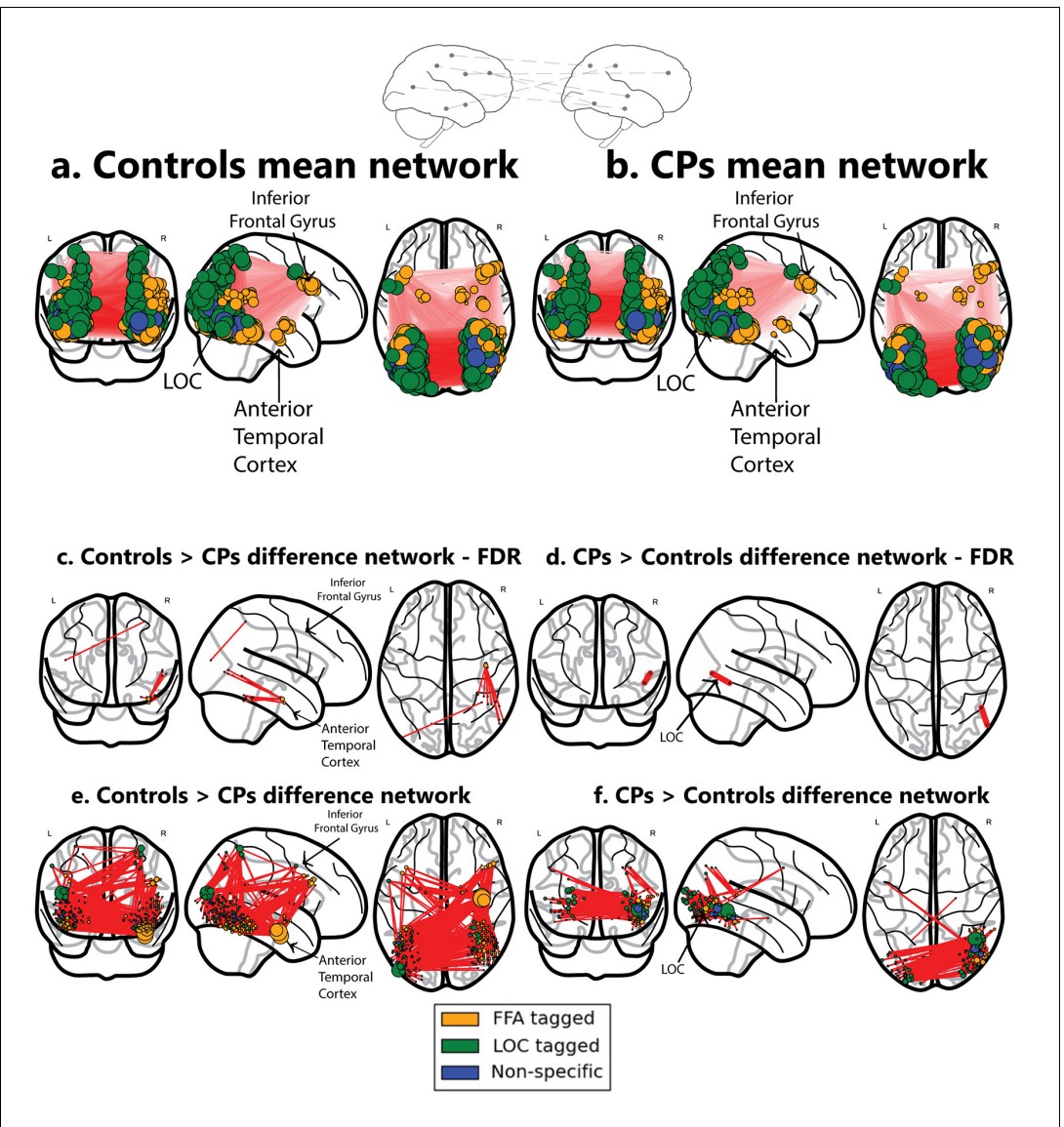

**Figure 1.** Networks obtained using the ISFC approach.  The top schematic denotes the inter-subject functional correlation approach, which calculates the inter-regional correlations in the brains of different individuals. (**a**) Mean network obtained from the control group (**b**) Mean network obtained from the CP group. These mean networks are shown for visualization purposes. For each group, networks are projected on three views of the brain (coronal, sagittal and axial views). The colors of the nodes denote their functional selectivity (face-selective, non-face selective, and nodes which are not exclusively selective for either of these stimuli). For visualization purposes, the size of the node is proportional to its degree (the larger the node, the greater its ISFC). The same conventions are used in all figures. (**c**) The difference network of controls > CPs after FDR correction (q < 0.05) which reflects the difference between the ISFC values for the controls compared to the CP individuals. The ATL and the inferior frontal gyrus are marked. As is graphically depicted, the ATL serves as the main hub for controls but not for CP. Three nodes, which comprise the ATL, were ranked in the top 10 ranks of degree scores (1-3; see Table 3a). (**d**) Difference networks obtained from the comparison of CPs > controls. CPs evince a significant difference in ISFC in posterior visual regions. As the FDR correction is very stringent, the differences are also shown for an uncorrected p<0.005 value. Note that this is an arbitrary threshold and was chosen for visualization purposes. (**e**) For the controls > CPs comparison, ATL serves as the main hub. (**f**) For the CPs > controls, hyperconnectivity of the network in posterior regions is apparent. For an interactive visualizations of the two networks from controls > CPs and CPs > controls, see https://goo.gl/mCm3OF and https://goo.gl/nctuOb respectively (Dwyer et al., 2015).

and CFMT scores of each participant (see Materials and methods for details). We hypothesized that higher ISFC values in the network edges located in posterior visual areas (e.g. LOC) should suffice for face representations that are largely driven by the immediate visual input and do not necessarily enable recognition (see *Gainotti, 2007*, *2013*; *Fox et al., 2008* and the Discussion section).

We focused the analysis on the single, significant posterior edge that was obtained from the contrast of the CPs>controls and examined the raw ISFC value of this edge within the CP and control groups. Moreover, based on previous studies (*Furl et al., 2011*), we set to examine whether the two behavioral measures are redundant and consequently, whether there is a latent behavioral measure which can account for both. In accordance with this study, we found a significant correlation between the CFMT and famous faces behavioral measures $r(18)=0.71$, p=0.0003. Accordingly, the 2 measures were factorized using PCA (*Furl et al., 2011*), and the first principal component which accounted for 85.9% of the variance served as our new face recognition behavioral score. The correlation between the face behavioral score was $r(18) = 0.9$, p<0.0001 and $r(18) = 0.94$, p<0.0001 for CFMT and famous faces questionnaire respectively. Subsequently, a multivariate regression was conducted to examine if ISFC and group predicted the face recognition behavioral score. This was done for the significant ISFC edges obtained from both the CP >Controls and the Controls > CP contrasts. As for the CP >controls contrast, overall a significant effect was found for both the group and the ISFC edge and the independent variables indeed explained a significant amount of the variance of the face recognition behavioral score ($R^2 = 0.83$, $R^2_{Adjusted} = 0.81$, $F(2,17) = 42.67$, p<0.0001). Moreover, the group significantly predicted face recognition behavioral score ($\beta = -6.08$, p<0.0001, $t = -6.83$, stde = 0.87), as did the ISFC value ($\beta = 4.89$, p=0.017, $t = 2.63$, stde = 1.85). The intercept term was not statistically significant ($\alpha = 0.93$, p=0.089, $t = 1.8$, stde = 0.52). Hence, ISFC score was positively associated with behavioral face performance (see *Figure 2a*). When conducting the same procedure but examining the edges obtained from the controls>CPs contrast, the ISFC edge and the independent variables also explained a significant amount of the variance of the face recognition behavioral score ($R^2 = 0.77$, $R^2_{Adjusted} = 0.74$, $F(2,17) = 29.19$, p<0.0001). Nevertheless, only the group effect coefficient was significant ($\beta = -5.23$, p=0.005, $t = -3.22$, stde = 1.62), while the ISFC coefficient was not ($\beta = -5.12$, p=0.44, $t = -0.79$, stde = 6.48). The intercept term was not significant, but showed a strong trend ($\alpha = 3.17$, p=0.052, $t = 2.09$, stde = 1.51). Thus, indicating the lack of a linear relation between the behavioral measure and ISFC in this contrast.

## Network analysis

In the previous section, we used the stringent FDR correction. To further understand and quantify the networks obtained from comparing CPs and controls, we used a complementary approach in which the non-thresholded positive t-values matrix connectivity patterns were compared to various topological and topographical attributes, while using correspondingly appropriate weighted graph measures for each contrast (CP>control and control>CP). These group differences were further quantified for each contrast using a measure of 'node strength', which quantifies the sum of positive edge weights connected to a node (*Rubinov and Sporns, 2010*). First, we assessed the selectivity of the differences in the ATL connectivity. The ISFC of the edges connecting the ATL to both face and non-face selective nodes located throughout the visual cortex was higher in controls>CPs compared to the opposite contrast (see specific pattern of ATL ISFC in *Table 2*) (*Figure 1e,f*). The strength scores of all nodes in the network were ranked in a descending order, and, of interest, the three nodes located in the ATL were ranked as the top 3 for the control subjects (see *Figure 1e,f* and *Table 3a*). Note that the ATL served as a hub connecting both face and non-face selective nodes (see *Table 2*). The node rankings confirm the centrality of the ATL in the face network of controls in contrast with that of CP, whose top 3 rankings include the right inferior temporal gyrus and the left lateral occipital cortex.

The network obtained from the CPs>controls comparison revealed that, at a low-edge threshold, significant edges were located throughout the visual cortex, but, as the threshold was increased, edges from anterior regions were eliminated and the remaining significant edges were located only in posterior parts of the visual cortex (*Figure 3*). To further quantify this effect, the Y coordinates of the 3D MNI space were ranked in an ascending order and binned into 10–21 equally sized bins measuring distance parallel to the posterior-anterior commissure axis (the maximal number of bins was chosen with a constraint that each bin contained at least one node). The significance level of the posterior to anterior pattern was then quantified using the Spearman correlation between the Y

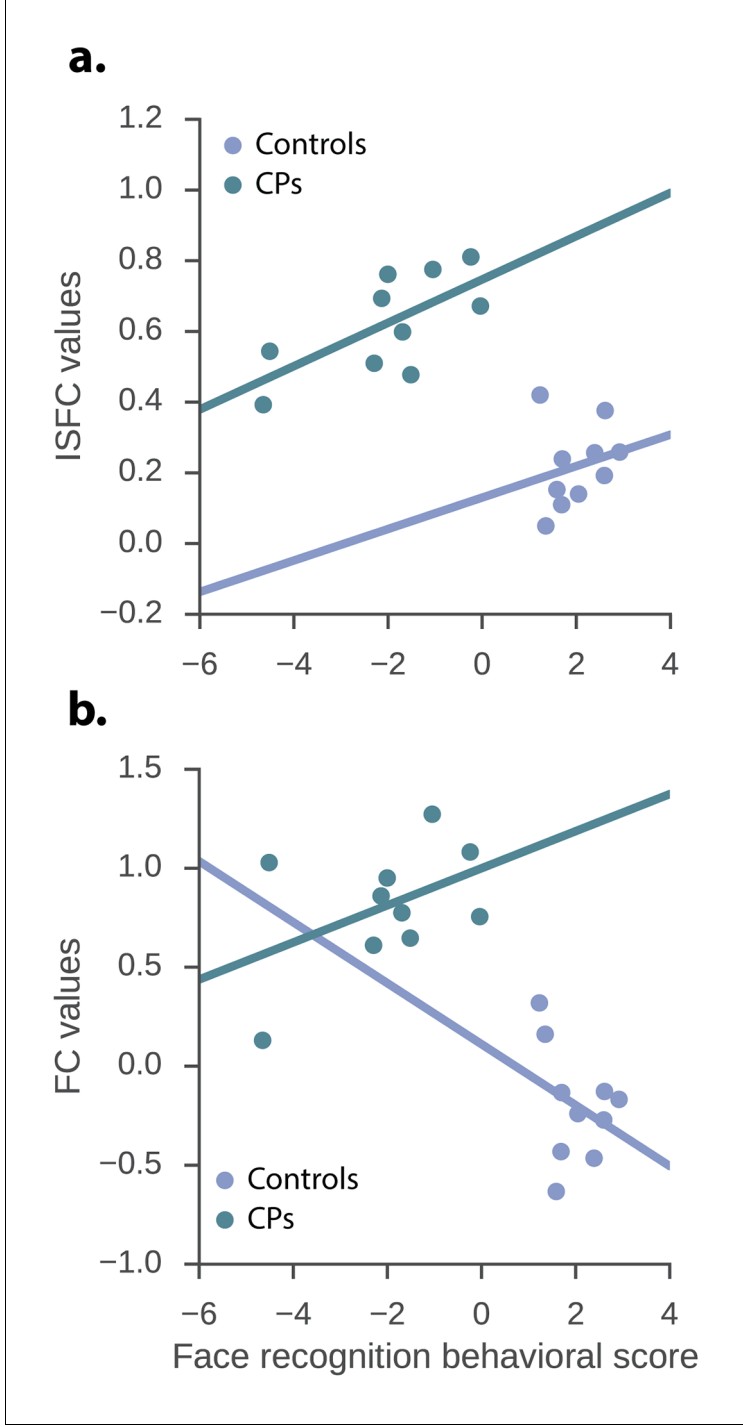

**Figure 2.** The relation between the neuronal measures and the face recognition behavioral score. ISFC and FC values of the significant CPs > controls edges and the face recognition behavioral score. (a) *ISFC regression –* group, as well as ISFC value significantly predict face recognition behavioral score (group: $\beta = -6.08$, p<0.0001, $t = -6.83$, std = 0.87; ISFC value: $\beta = 4.89$, p=0.017, $t = 2.63$, std = 1.85). (b) *FC regression -* group significantly predict face recognition behavioral score while the FC value does not (group: $\beta = -4.99$, p<0.0001, $t = -4.68$, stde = 1.06; FC value: $\beta = 0.95$, p=0.31, $t = 1.04$, stde = 0.91).

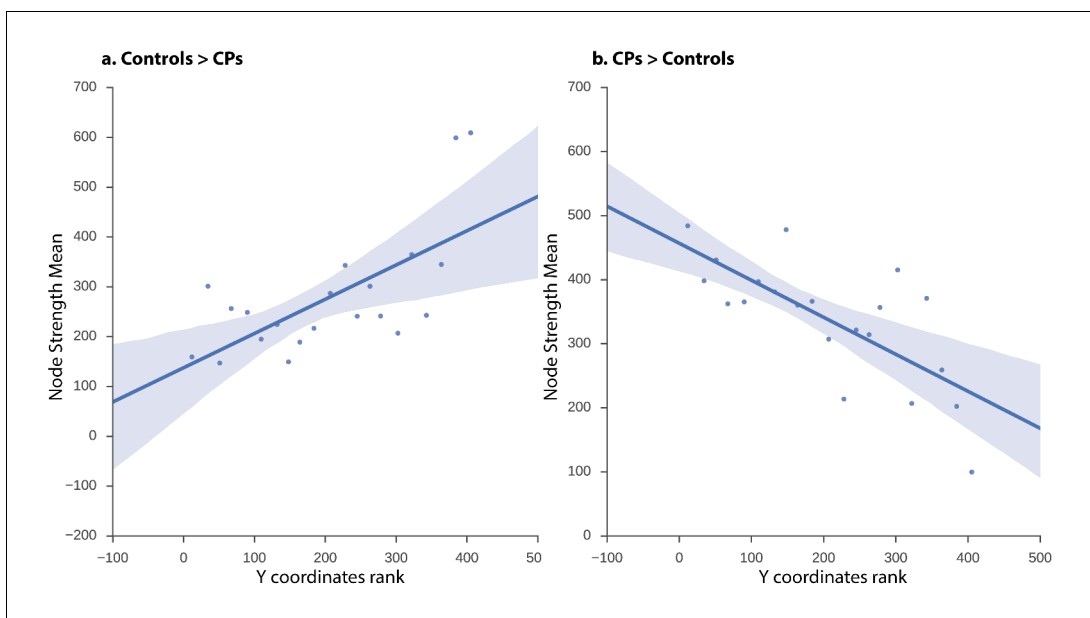

**Figure 3.** ISFC correlation between strength and node location along posterior-anterior axis. Linear regression fit with a 95% confidence interval band between the strength measure of the nodes and the Y coordinate ascending rank order of each node binned into 21 equally sized bins for (a) controls > CPs ($r_s$ = 0.58, p=0.005) and (b) CPs > controls (rs = −0.70, p=0.0003). The x-axis of the graph denotes the Y coordinates of the 3D MNI space ranked in ascending order, the y-axis of the graph marks the mean strength value of nodes per bin. As is evident, the higher the y-coordinate (more anterior), the higher the strength value for controls > CPs and the lower the value for CPs > controls.

coordinate bins and the nodes' strength for CPs>controls and controls>CPs contrasts separately. The results indicate that the higher Y coordinates (more anterior regions along the anterior posterior axis) were associated with higher positive strength values for the controls>CPs contrast across all bin divisions ($r_s$ = 0.52 to 0.8, p=0.004 to 0.053). Note that division into 14 bins resulted in the lowest correlation ($r_s$ = 0.52), which was only marginally significant (p=0.053). Contrarily, the CPs>controls contrast revealed an opposite result such that more posterior regions had higher node strengths across all bin divisions ($r_s$ = −0.67 to −0.83, p=0.0003 to 0.01). These correlations validate the result of greater posterior ISFC in CPs versus controls.

Additionally, this analysis revealed face-specific dominance, such that the nodes that had the highest degree were face-selective. Specifically, higher ISFC patterns in the control group, compared to the CP group, were associated with face selective nodes. For this comparison, eight out of the top ten nodes (80%) were face-related (ranked in descending order by degree score) (*Table 3a*), compared to an overall face node base rate of 35% (number of face nodes divided by overall number of nodes). The difference between the face-selective node rate in the controls > CPs contrast and the overall face nodes base rate was statistically significant, $\chi^2$(1, n = 20) =5.05, p=0.02. Moreover, 9 out of these 10 nodes, consisted of voxels originating from the same seed (FFA or LOC). In the remaining node, voxels consisted of 90% of one tag (either FFA or LOC) and 10% of the other tag. Thus, the face network of the controls was associated with a higher number of face-tagged nodes compared with the face network of the CPs. When comparing the opposite contrast (CPs>controls), no statistically significant difference was found in the face-selectivity of the nodes $\chi^2$(1, n = 20) =1.25, p=n .s. In support of these findings, 7 out of the 10 nodes included a substantial majority of LOC tagged voxels (>75%). As for the remaining 3, they were more heterogeneous and contained a mixture of voxels with all 3 possible taggings. In fact, four of the top ten nodes, of which only one was face specific, belonged to the lateral occipital cortex (one in the left hemisphere and three in the right) and six nodes belonged to the adjacent right inferior temporal cortex.

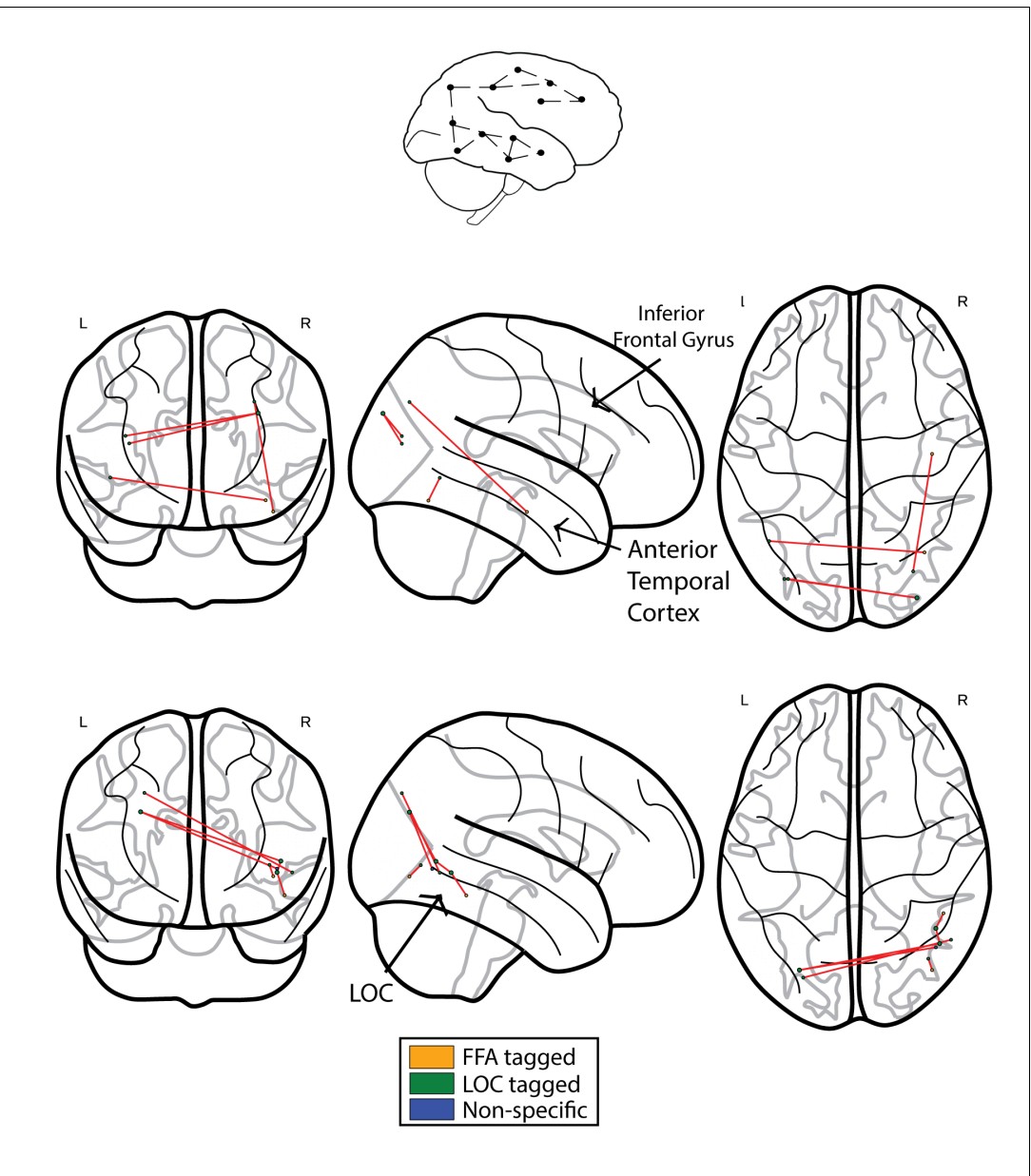

**Figure 4.** Networks obtained using the FC approach. The top schematic denotes the intra-subject functional correlation, which calculates the inter-regional correlations within the brains of single individuals. (a) The FC difference network of controls>CPs. This comparison reflects the difference between the FC correlation coefficient values of the controls compared to the group of CP individuals. The maps are presented following the application of the FDR correction (q < 0.05). (b) Difference networks obtained from the comparison of FC in CPs>controls.

## Functional correlation analysis

We next compared the results obtained using ISFC to the results obtained using standard functional connectivity (FC) analysis. In accordance with our prediction, when comparing the controls to CPs, we observed the expected greater connectivity between the ATL and posterior regions. This effect was evident in a single edge following the application of the FDR multiple comparisons procedure (see *Figure 4a*, controls>CPs). Similar to the ISFC, the opposite contrast of CPs>controls revealed hyperconnectivity in posterior visual regions with a main hub in the vicinity of the LOC and inferior temporal cortex. Critically, this effect was also evident following FDR correction (see *Figure 4b*, CPs>controls).

## Correlations of the significant edges with behavior

The multiple regression analysis described above was repeated. Specifically, we used the edges obtained from the FC as opposed to the ISFC and group independent variables to predict the face recognition behavioral score. This was done for the significant FC edges obtained from both the CP>Controls and the Controls>CP contrasts (*Figure 2b*). The regression model with FC and the group independent variables explained a significant amount of the variance of the face recognition behavioral score for both CPs>controls ($R^2 = 0.78$, $R^2_{Adjusted} = 0.75$, $F(2,17) = 30.17$, $p<0.0001$) and controls>CPs ($R^2 = 0.78$, $R^2_{Adjusted} = 0.75$, $F(2,17) =30.78$ , $p<0.0001$). Nevertheless, for both the CPs>Controls and Controls>CPs only the group effect coefficient was significant ($\beta = -4.99$, $p<0.0001$, $t = -4.68$, stde = 1.06) and ($\beta = -5.23$, $p<0.0001$, $t = -4.52$, stde = 1.15) respectively, while the FC coefficient was not ($\beta = 0.95$, p=0.31, $t = 1.04$, stde = 0.91) and ($\beta = -1.34$, p=0.25, $t = -1.16$, stde = 1.15) respectively. Moreover, the intercept terms were statistically significant ($\alpha = 2.2$, p<0.0001, $t = 5.34$, stde = 0.41) and ($\alpha = 2.77$, p=0.002, $t = 3.71$, stde = 0.74) for the CP>Controls and Controls>CPs contrasts respectively. Thus, no direct link was found between the face recognition behavioral score and FC connectivity. Hence, in accordance with (*Simony et al., 2016*) only the ISFC, but not standard FC could be linked to the behavioral measures thus capturing an additional important aspect of the data.

## Estimation of the signal to noise ratio of nodes

The stimulus locked response patterns among nodes in the posterior occipital cortex were highly reliable among the CP group, as measured by ISFC. Such stimulus locked reliable activation suggests that CP individuals process visual information in a unique, but systematic fashion, which is not observed in the control group. Such positive effect rules out the possibility that the differences in network topology across the groups emerge from a global reduction in SNR within the CP subjects. Rather, this finding indicates that the stimulus-locked activity patterns are reliable in both groups, but that the network topology differed across the two groups.

# Discussion

The study of individuals with CP is useful from both a basic science and a translational perspective. On the one hand, CP individuals provide a unique opportunity to examine the underlying network function of normal face processing while avoiding some of the pitfalls that arise when studying patients with frank lesions (and altered vasculature) to cortex and, on the other hand, characterizing the deficits in these individuals relative to controls elucidates the atypical profile of this neurodevelopmental disorder. Also, because face recognition appears to be accomplished via the coordinated activity of multiple nodes of a distributed neural network (*Haxby et al., 2000*; *Fairhall and Ishai, 2007*; *Davies-Thompson and Andrews, 2012*; *Gschwind et al., 2012*; *Joseph et al., 2012*; *Phillips et al., 2012*; *Pyles et al., 2013*; *Zhen et al., 2013*), such a study provides the opportunity to determine whether a disrupted function of one or more of the nodes within this network leads to an altered organization of the network and, if so, to document the nature and topology of this altered organization.

By isolating stimulus-locked neural responses using a novel ISFC analysis, the present study revealed alterations in the topological structure of the visual network in CP vs control subjects.

We exploited an ISFC approach, which filters out intrinsic neural dynamics and non-neuronal artifacts that can influence FC patterns within a brain but are uncorrelated across brains of different participants (*Hasson et al., 2009*; *Simony et al., 2016*). We aimed to characterize and contrast the normally functioning face network in healthy individuals with the face network of individuals with CP who are markedly impaired at face processing. The ISFC analysis revealed a unique pattern of anterior to posterior differences in the significance of nodes in the network of CPs vs. controls. Specifically, we documented hyperconnectivity in posterior visual areas in CPs vs. controls, and hypoconnectivity between the occipital areas and anterior temporal and frontal areas. Such alterations in the topology of the correlation patterns were correlated with behavioral scores of participants in standard measures of face processing. A previous study has shown that the ISFC better captured momentary network configurations and better predicted behavioral performance (*Simony et al., 2016*). Consistently with this study, in the context of the present study, only the

ISFC, but not FC was shown to be related to the behavioral measure of face processing, and hence was able to elucidate an additional important aspect of the data. In addition, a complementary analysis revealed face-specific dominance, reflected as higher degree nodes that are face selective (defined by a localizer), in the control group compared to the CP group. In fact, in the comparison of CPs>controls, six of the top ten nodes belonged to the right inferior temporal gyrus and three nodes belonged to the lateral occipital cortex (LOC), of which only a single node was face specific.

## The anterior temporal cortex

Perhaps the most striking difference in the topology of the face network in the controls versus CPs concerns the differences in the connectivity to the anterior temporal lobe (ATL). The ATL is implicated in the integration of person-specific information (*Kriegeskorte et al., 2007*; *Simmons et al., 2010*; *Nestor et al., 2011*; *Yang et al., 2016*) and familiar people recognition (*Gainotti et al., 2003*; *Gainotti, 2007*). Specifically, damage to the left ATL is more associated with impaired representation of semantic information, while damage to the right ATL impedes the visual recognition of familiar faces (*Gainotti, 2007*). Based on both functional activation data, functional connectivity data (during task and at rest) and structural volumetric and connectivity findings, we have argued previously that an abnormality in the anterior temporal lobe and its connectivity may play a critical role in the neural basis of CP (*Behrmann et al., 2007*; *Avidan and Behrmann, 2009*; *Avidan et al., 2014*; *Thomas et al., 2009*). The abnormality of the ATL is evident in the current findings as well, and the analysis, conducted in an assumption-free fashion revealed that the ATL was the most important hub that distinguished the network topology of the CPs and controls. Together, these findings point to abnormal structure, function and connectivity of this region in CP individuals (but see *Gomez et al., 2015*; *Song et al., 2015*; *Lohse et al., 2016*).

## Altered organization of the face network in CPs

Our findings indicate that the impaired ISFC of the ATL may result in hyper-ISFC of the more posterior nodes, as in the right inferior temporal gyrus and the LOC, in CP compared to controls. These alterations in topology offer a possible account for the face recognition deficits exhibited by CP individuals (*Avidan et al., 2011*; *Tanzer et al., 2013*; *Yang et al., 2016*). For example, the posterior hyper-connectivity in CP, in tandem with the residual, weak, connectivity with anterior regions may allow for some structural encoding of face stimuli derived from the immediate visual input (see *Behrmann et al., 2005*) for relevant behavioral findings). Although insufficient for facial recognition and individuation of identity, this rudimentary processing may partially compensate for the failure to utilize the ATL and its connectivity (*Yang et al., 2016*).

Furthermore, in the current study, CP subjects exhibited greater LOC-related ISFC compared to normal subjects. Consistent with this finding, it has been demonstrated that individuals with autism spectrum disorder (ASD) who have selective deficits in face recognition show greater activation of the LOC (*Schultz et al., 2000*; *Hubl et al., 2003*). Numerous studies have shown that the LOC is associated primarily with object perception (*Malach et al., 1995*; *Grill-Spector et al., 2001*; *Freud et al., 2015*) and plays a major role in the processing of inverted faces, perhaps through object-like and feature-based processing (*Rosenthal et al., 2016*; *Yovel and Kanwisher, 2005*; *Pitcher et al., 2011*; *Matsuyoshi et al., 2015*).

Together, these findings offer a possible account for the fact that most CP individuals, despite their severe deficit in recognizing the identity of individual faces, are typically able to detect the presence of a face in a scene. Specifically, it is possible that the network edges located in posterior visual areas (e.g. LOC) allow for the computation of face representations that are largely driven by the immediate visual input. This information may be relied on disproportionately by the CP individuals and perhaps serves as partial compensation for the failure to represent person-selective information which, in the normal brain is supported by the anterior temporal cortex and its connectivity (*Gainotti, 2007*, *2013*; *Fox et al., 2008*). The current investigation does not allow us to infer causality, and thus, it remains to be determined whether the disconnection between the anterior and the posterior regions are the cause or the effect of the altered network organization.

A final related, but unresolved issue, is whether the impairment in CP represents the lower end of the continuum of the normal distribution of face processing abilities, or whether the face recognition deficit in CP reflects a separate phenomenon and, hence, a distinct disorder. This important issue is

outside the scope of the present study and further imaging and behavioral research is warranted in order to address it.

## Conclusions

To the best of our knowledge, these data provide the first demonstration of wide-scale network topological differences in individuals with CP. Utilizing the ISFC approach, the results validated previous atypical ATL connectivity findings in CP and enabled us to gain further insight into the altered network-wise configuration of individuals with CP. We propose that investigations of the topology of other neurodevelopmental disorders might benefit from the analytic approach developed here and that insights into their underlying neural mechanisms might also be gained (*Steinbrink et al., 2008*; *Odegard et al., 2009*).

# Materials and methods

A requirement for network analysis is the initial formulation of components of the network as a set of nodes and edges, and there are many ways in which this can be accomplished (*Bullmore and Sporns, 2009*; *Rubinov and Sporns, 2010*). Below we detail the specific approach we have taken:

## Definition of the network edges

A standard measure which is often used in fMRI studies for characterizing the edges in a functional network is the correlation coefficient between pairs of nodes identified within each subject (*Smith et al., 2011*). This approach was applied in the present study as well. However, despite its popularity, a major limitation of this measure, which captures within-subject synchronous activity, often referred to as functional connectivity, is that spontaneous intrinsic neuronal activity cannot be reliably separated from the evoked activity associated with the task (*Greicius et al., 2003*; *Fox and Raichle, 2007*; *Simony et al., 2016*). Additionally, non-neuronal artifacts such as respiratory rate and motion might also affect the results, usually by decreasing the signal to noise ratio (SNR).

A possible partial solution to these limitations, formulated to increase the network inference SNR (*Simony et al., 2016*), relies on network construction using inter-subject functional correlation (ISFC) rather than exploiting the more widely used within-subject functional correlations. Briefly, for each pair of preselected regions of interest (ROIs) or nodes, correlation coefficients are computed between each participant and the mean signal of the remaining group without this particular participant (leave-one-out), and then correlations are averaged across all pairwise iterations. As the correlations are calculated between different participants, any intrinsic activity and artifacts, which are not correlated between participants, are cancelled out and, consequently, only the task-induced activity of interest serves as the basis of the correlations, resulting in an improved SNR.

The present study is conducted on individuals with CP, a relatively uncommon disorder and, hence, sample size is inherently limited, especially when contrasting the group size against the large number of nodes and edges (i.e., the number of variables outweighs the number of samples). This situation is typical in investigations of special populations and so we used a version of the inter-subject functional correlation (ISFC) (*Hasson et al., 2009*; *Simony et al., 2016*) to increase the SNR.

## ISFC formulation

ISFC, is composed of the following steps: First, each experimental group (e.g. controls and CPs) which is composed of 10 subjects is divided into a group of 9 subjects and one remaining subject, and the raw signal for each node is averaged across all participants in the 9 subjects group. Second, for each experimental group, correlation coefficients are calculated between each pair of independent nodes, such that each pair of nodes is composed of one node from the individual subject and one node from the mean of the remaining nine subjects. The correlation coefficients are transformed by Fisher z-transformation. This process is repeated for all subjects in each experimental group such that, in each iteration, one of the subjects is left out. At the end of the process, each subject in each experimental group has all correlation coefficient values between all its nodes and the mean of all the other 9 subjects.

An independent-samples t-test is conducted for each pair of nodes to compare its ISFC in CP and control. The positive and negative t-values reflect the controls>CPs and CPs>controls contrast respectively.

Next, to determine statistical significance and assess the null distribution, the subjects are randomly divided into 2 groups of 10 and the same procedure is performed on permuted groups 10000 times. This results in a null distribution for each edge. Finally, an empirical p-value is calculated for controls>CPs and for CPs>controls as the number of times that each empirical t-value was smaller compared to the null distribution of each edge. The p-values are corrected for multiple comparisons using the false discovery rate [qFDR<0.005; (*Benjamini and Hochberg, 1995*)] procedure.

## A priori localizer for node definition

### Participants and ethical approval

Sixteen healthy, right-handed individuals (8 females) with normal or corrected-to-normal vision participated in the experiment (mean age ±SD = 24.5±1.11). The data from three additional participants were discarded due to excessive noise. The experiment was approved by the Helsinki committee of the Soroka University Medical Center, Beer Sheva, Israel (5106) and all participants provided signed informed consent prior to participation.

### MRI setup

Participants were scanned in a 3T Philips Ingenia scanner equipped with a standard head coil, located at the Soroka Medical Center, Beer Sheva, Israel. fMRI BOLD contrast was acquired using the gradient-echo echo-planner imaging sequence with parallel acquisition (SENSE: factor 2.8). Specific scanning parameters were as follows: whole brain coverage 35 slices, transverse orientation, 3 mm thickness, no gap, TR = 2000 ms, TE = 35 ms, flip angle = 90°, FOV = 256 × 256 and matrix size 96 × 96. High-resolution anatomical volumes were acquired with a T1-weighted 3D pulse sequence (1 × 1 × 1 mm3, 170 slices).

### Visual stimulation

Stimuli were presented using the E-prime 2.0 software (Psychology Software Tools, Inc., Pittsburgh, PA, USA) and projected onto an LCD screen located in the back of the scanner bore behind the subject's head. Participants viewed the stimuli through a tilted mirror mounted above their eyes on the head coil.

### Localizer scan

A standard blocked-design localizer experiment was used to define face and non-face selective regions. Stimuli were presented in 10 s blocks of famous faces, unfamiliar faces, buildings, daily objects, or scrambled objects (1 image was presented twice as part of the task) interleaved by 6 s rest periods. Within each block there were 9 images. Each image was presented for 800 ms followed by 200 ms inter-stimulus interval and participants performed a two alternative forced choice task (see detailed description of the protocol in [*Avidan et al., 2014*]). The data from these participants were used to identify the nodes or regions to be used in the analysis of the CP individuals.

## Main experimental scans

### Participants and ethical approval

All participants had normal or corrected-to-normal vision. The experiment was approved by the Institutional Review Boards of Carnegie Mellon University (IRBSTUDY2015_00000191), and all participants provided signed informed consent..

### Congenital prosopagnosia

Ten healthy [8 right-handed as confirmed by the Edinburgh Handedness inventory (*Oldfield, 1971*)] individuals diagnosed with CP (8 females, 2 males), aged between 18 and 62 years, participated in this study (mean age ±SD = 40.04 ± 15.03). None of the CP individuals had any discernible lesion on conventional MRI scanning, and none had a history of any neurological or psychiatric disease by self-report. All CP participants reported substantial lifelong difficulties with face processing. The data for 7 of the 10 CPs have been reported previously (see detailed description in *Table 1*). Detailed behavioral profiles were obtained and only those participants whose performance fell below 2 standard deviations of the mean of the control group on at least 2 of the 4 diagnostic measures were included [Cambridge Face Memory Test (CFMT), Famous faces questionnaire, Cambridge Face Perception

**Table 1.** CP behavioral scores ordered by severity as indicated by performance on the CFMT.

| Participant | Sex | Age | Famous faces questionnaire | | CFMT (total) | |
| --- | --- | --- | --- | --- | --- | --- |
| | | | % corr. | Z- score | | |
| BL | F | 18 | 20 | -4.88 | 28 | -4.15 |
| BQ | F | 29 | 16 | -4.92 | 30 | -3.89 |
| KG | F | 49 | 75 | -0.69 | 33 | -3.5 |
| ON (*Avidan et al., 2014*) | F | 48 | 60.7 | -1.77 | 35 | -3.23 |
| MT (*Nishimura et al., 2010*), (*Avidan et al., 2011*), (*Behrmann and Avidan, 2005*), (*Thomas et al., 2009*), (*Avidan and Behrmann, 2008*), (*Humphreys et al., 2007*) , (*Behrmann et al., 2007*), (*Avidan et al., 2014*) | M | 50 | 62.5 | -1.64 | 36 | -3.11 |
| WA (*Nishimura et al., 2010*), (*Avidan et al., 2014*) | F | 23 | 45.7 | -2.91 | 40 | -2.58 |
| KE (*Avidan et al., 2011*), (*Thomas et al., 2009*), (*Avidan et al., 2014*) | F | 67 | 42.9 | -3.12 | 40 | -2.58 |
| TD (*Nishimura et al., 2010*), (*Avidan et al., 2011*), (*Avidan et al., 2014*) | F | 38 | 46.4 | -2.85 | 41 | -2.45 |
| MN (*Nishimura et al., 2010*), (*Avidan et al., 2014*) | F | 50 | 60.7 | -1.77 | 52 | -1 |
| BT (*Avidan et al., 2011*), (*Avidan et al., 2014*) | M | 32 | 55.3 | -2.18 | 58 | -0.21 |
| CP Mean ± s.d | | 40.4± 15.03 | 48.52 ± 18.72 | | 39.3 ± 9.41 | |
| Control Mean ± s.d | | 39.3 ± 13.4 | 91.57±6.24 | | 58.28 ± 5.87 | |

The table shows the age and gender of participants and their performance (raw values and z-normalized scores relative to a large control group) on the famous faces questionnaire and CFMT. Note that 7 of the 10 CPs have participated in previous behavioral (3 CPs), and imaging (7 CPs) studies; additional behavioral measures for the CP individuals can be found in these references. Specific details regarding diagnostic and inclusion criteria can be found in the Materials and methods section and in the related studies. The average performance on the famous faces questionnaire and the CFMT of the controls who participated in the present study is also provided (t-test comparing performance across the CP and the controls, for famous faces questionnaire p<0.0005 and CFMT p<0.0005).

Test (CFPT), and a task measuring discrimination of novel upright and inverted faces; see description of the behavioral tests and details regarding prior publications for each subject in *Table 1*.

## Matched controls
Ten healthy individuals, aged 25–62 years (mean age ±SD = 39.3±13.4), who did not report having any difficulties with face processing participated in the imaging experiment. The CP and age-matched controls did not differ in age (p=0.84). There was a significant difference between the control subjects and the CP group on their performance in the famous faces questionnaire (t(15) = 6.09, p<0.0005) and CFMT (t(15)= 5.54, p<0.0005; see *Table 1* for mean performance), confirming the behavioral deficit in the CPs included in this study.

## Imaging experiment
### Visual stimulation
Visual stimuli were generated using the E-prime IFIS software (Psychology Software Tools, Inc., Pittsburgh, PA, USA) (for details see [*Avidan et al., 2014*]).

### MRI setup
Subjects were scanned either in a 3T Siemens Allegra scanner, equipped with a standard head coil (5 CPs, 4 controls) or in a 3T Siemens Verio scanner equipped with a standard head coil (5 CPs, 6 controls), using similar scanning parameters. For detailed description of the specific scanning parameters and acquisition order during the scanning session see (*Avidan et al., 2014*).

### Visual stimulation experiment
Stimuli consisted of 10 images of emotional faces (angry, fearful), neutral faces, famous faces, unfamiliar faces (*Avidan and Behrmann, 2008*) or buildings, presented in separate 10 s blocks. Blocks

**Table 2.** Coordinates of the ATL nodes, and the calculated values of the within module weighted degree and participation coefficient.

| Number Of Voxels | MNI center coordinates | Within module weighted degree | Participation coefficient |
|---|---|---|---|
| 23 | 42, -14, -26 | 3.91 | 0.58 |
| 13 | 40, -10, -32 | 4.81 | 0.58 |
| 20 | 40, -12, -28 | 5.15 | 0.59 |
| Voxels | 40, -10, -32 | Weighted Mean: | Weighted Mean: |
| Sum: | | | |
| 55 | | 4.63 | 0.58 |

Coordinates of the ATL are in line with (***Rajimehr et al., 2009***; ***Pyles et al., 2013***). To assess whether the differences are specific to face-selective nodes, the ratio of face-selective nodes and non-face selective nodes connected to the ATL nodes was quantified using within-module weighted degree and participation coefficient. Within-module weighted degree measures the ISFC level of a node within its module. In this analysis, a module is defined as one of the three types of functional tags (faces, non-faces and overlap). Note that this definition is different from the graph theoretical modularity measure as used in the first section of the results. The participation coefficient measures the inter-module diversity of the nodes' connections, meaning how much a node is connected not only within its own module but across modules (***Guimerà and Nunes Amaral, 2005***).

This resulted in a within-module weighted degree weighted average of 4.63 and a participation coefficient weighted average of 0.58. Values greater than 2 are considered as a 'module hub'. Additionally, 'module hubs' with a participation coefficient between 0.3 and 0.75 are treated as 'connector hubs', that is hubs with many connections to most of the other modules (***Guimerà and Nunes Amaral, 2005***). Importantly, based on these standard values, the differences between controls and CPs, assigned to the ATL, are not a priori face specific.

were separated by 6 s rest periods and there were 7 repetitions of each block type. Each image was presented for 800 ms followed by 200 ms inter-stimulus interval. Participants performed a 2 alternative-forced choice task during scanning: they had to press a button on a MRI-compatible response box if two consecutive images were the same. Detailed description of the stimuli can be found in (***Avidan et al., 2014***).

## Data analysis
### Preprocessing of anatomical data
Anatomical scans were first preprocessed using the FSL (RRID:SCR_002823) anatomical processing script (fsl_anat) which includes the reorientation of the images to the standard (MNI) orientation (fslreorient2std), automatic cropping of the head from the image (robustfov), bias-field correction (RF/B1-inhomogeneity-correction) (FAST), registration to a 2 mm MNI standard space (linear and

**Table 3.** Rank of top 10 nodes obtained from the comparisons of CP and the control group.

| a. Controls>CPs | | b. CPs>Controls | |
|---|---|---|---|
| Rank | Region Name | Rank | Region Name |
| 1,2,3 | Anterior temporal Cortex (faces) | 1,3,4,5,9,10 | Right inferior temporal gyrus (non-faces) |
| 8 | Right Temporal Occipital Fusiform (faces) | 2 | Left lateral occipital cortex (non-faces) |
| 4 | Left TOS (non-faces) | 6 | Right lateral occipital cortex (non-faces) |
| 7 | Left Lateral Occipital cortex (non-faces) | 7 | Right lateral occipital cortex (faces) |
| 9,10 | Left inferior frontal gyrus (faces) | 8 | Right lateral occipital cortex (non-selective) |
| 5,6 | Amygdala left (faces) | | |

The 10 nodes with the highest node strength obtained from the Controls>CPs (a) and CPs>Controls (b) difference networks. Anatomical locations, which are based on 'The atlas of the human brain' (***Mai et al., 2008***) and validated by an expert, are provided for each node. Note that, in the controls>CP network, most of the nodes were face selective, while in the CP>controls network, only a single node was face selective.

non-linear) (FLIRT and FNIRT), brain-extraction (FNIRT-based), tissue-type segmentation (FAST) and subcortical structure segmentation (FIRST) (*Jenkinson et al., 2012*).

## Preprocessing of functional data

Preprocessing was conducted using dedicated Nipype pipeline v.011 (*Gorgolewski et al., 2011*; RRID:SCR_002502). We utilized different components from various neuroimaging software packages including SPM8 (*Penny et al., 2011*; RRID:SCR_007037), FreeSurfer v5.0 (*Fischl, 2012*; RRID:SCR_001847) and FSL 5.0 (*Jenkinson et al., 2012*; RRID:SCR_002823) as well as in-house code written in Python (code will be provided upon readers' request).

The preprocessing consisted of volume realignment of the functional scans (6 directions) to the mean EPI using SPM8 (*Friston et al., 1996*); artifact detection of functional scans bounded by GM mask which marked as outliers images with intensities greater than 3 standard errors from the mean intensity and images with normed composite motion differences between successive motion volumes of 1 (*Gorgolewski et al., 2011*); registration of functional to anatomical scans using FSL's BBR registration procedure [epi_reg; (*Greve and Fischl, 2009*; *Jenkinson et al., 2012*)]; regressing out motion of CSF and WM first 6 components of PCA outliers [compcorr; (*Behzadi et al., 2007*)]; detrending (removal of second order polynomials); normalization to non-linear MNI space using transformation matrices which were obtained from FSL's anatomical preprocessing script; and, finally, spatial smoothing (6 mm) using SPM8 (*Penny et al., 2011*). [For related pipeline analyses used in other studies, see (*Smallwood et al., 2013*; *Schaefer et al., 2014*)].

## Definition of nodes

See description of data acquisition and experiment in the 'A priori localizer for node definition' section above.

In order to conduct a network analysis, one needs to define a sufficiently large number of nodes that, in themselves, are relatively small. To gain a finer resolution of the network while maintaining the functional origin of each node, a seed-based correlation mask was constructed based on all voxels activated by seeds in the FFA and LOC located in the right hemisphere. These high-order visual regions were selected as seeds due to their well-documented roles in face (*Yovel and Kanwisher, 2004*, *2005*; *Mazard et al., 2006*; *Pinsk et al., 2009*; *James et al., 2013*; *Rosenthal et al., 2016*) and non-face processing respectively. Specifically, our underlying theoretical hypothesis was that CP would exhibit a more 'object like' processing style compared to controls while utilizing the right LOC (*Matsuyoshi et al., 2015*; *Rosenthal et al., 2016*).

A SPM8 random-effects group analysis was conducted, such that in a first-level analysis, contrasts between stimulus categories were calculated and these were then used in a second-level analysis with subject as a random factor. The analysis included high-pass filtering with a cutoff of 1/128 Hz and a first-order autocorrelation correction. Right FFA and right LOC seeds were initially marked based on the contrasts of faces>(houses and objects) and (houses and objects) >faces using xjview, an SPM add-on tool for defining ROIs, after FDR < 0.05 correction (*Cui et al., 2011*). The selected voxels (see below for selection procedure) were later sub-divided into small, spatially constrained clusters. This procedure preserved the original functional specificity of each voxel with 'functional tagging' marking the functional preference of each node.

Using a custom Python script, voxels obtained from the functional localizer scans were limited to a gray matter mask based on the intersection of all the individual gray matter masks in MNI space (*Abraham et al., 2014*). Then, the time course was z-scored and a mean time course for all localizer runs averaged across all subjects was created. Using the mean time course, two separate seed-based correlation maps based on the right FFA and LOC initial seeds were defined and binarized using a 0.5 r value threshold. Our main motivation for choosing the 0.5 threshold was that it covered large portions of the cortex. A union between the two maps was created and a clustering analysis was performed using the scikit-learn (v0.18.1) Ward Agglomeration algorithm while utilizing a spatial neighborhood matrix to constrain clusters to form contiguous parcels on the nodes' mean time courses (*Pedregosa et al., 2011*) resulting in nodes which are either FFA-tagged, LOC-tagged, and nodes which are not exclusively tagged (i.e. voxels which were correlated to both the FFA and LOC seeds at r > 0.5). For each node, we counted the number of voxels associated with each functional tagging and the node's selectivity was determined by the mode. The upper bound of spatially

constrained sub-regions was set to 500 and was determined by practical reasons with a compromise between the need for a finely grained resolution of nodes, on the one hand, and the requirement to avoid regions that are too small and might result in poor signal to noise ratio, on the other. Hence, nodes with fewer than 10 voxels were eliminated from the analysis, resulting in 415 nodes for the obtained networks. The total number of voxels which were removed due to this threshold of a minimum cluster size was 313 voxels out of 13407 (eliminated nodes had 3.86 voxels on average with a 2.42 standard deviation, while remaining nodes had $31.55 \pm 18.4$ voxels on average). Of note is that one possible outcome of this unsupervised clustering analysis is that more than a single node can reside within the 'classical' functional definition (contrast based definitions) of a node such as for example, within the LOC or the FFA. This is evident in the Results section when the characteristics of the nodes are specifically described. Visualization of networks was done with custom Python script utilizing Nilearn library v0.2.5.1 (*Abraham et al., 2014*; RRID:SCR_001362).

## Definition of edges
### Standard functional connectivity (FC)

For each subject and localizer run, the time course of each of the nodes was extracted after standardization (zero mean and unit variance) and high-pass filtered using a cutoff of 1/128 Hz. The two localizer runs were averaged for each subject. Correlation matrices were constructed for each subject using pairwise Pearson correlation coefficients between each pair of nodes and a Fisher r-to-z-transformation was applied to each edge. Additionally, a z normalization was applied across all correlation coefficients for each subject to remove any subject level global effects. An independent-samples t-test was conducted to compare each edge in the CP and control group while correcting for FDR (q < 0.05). The outcome of this procedure are two networks which capture the significant differences in FC between controls > CPs and the reverse, CPs>controls.

## ISFC
### Split group analysis based to the main experimental scans

For each subject and localizer run, the time course of each of the nodes defined in the separate localizer scan was extracted after standardization (zero mean and unit variance) and high-pass filtered using a cutoff of 1/128 Hz. The two localizer scans were averaged. ISFC was performed on the average time course while contrasting the CP and matched controls group.

### Construction of difference networks

Two difference networks which capture the significant difference in ISFC between controls>CPs and CPs>controls were constructed. This was done for each of these comparisons separately in the following manner: Using the ISFC procedure, any edge that was not significantly different between the two groups, while correcting for FDR, was removed from the analysis and was set to zero. (See *Figure 1*).

## Acknowledgements

This work was support by ISF grant 296/15 to GA and NSF grants BCS-1354350 and #SBE-0542013 to MB.

## Additional information

### Funding

| Funder | Grant reference number | Author |
| --- | --- | --- |
| Israel Science Foundation | 296/15 | Galia Avidan |
| National Science Foundation | BCS-1354350 | Marlene Behrmann |
| National Science Foundation | #SBE-0542013 | Marlene Behrmann |

The funders had no role in study design, data collection and interpretation, or the decision to submit the work for publication.

## Author contributions
GR, Conceptualization, Data curation, Formal analysis, Investigation, Visualization, Methodology, Writing—original draft, Project administration, Writing—review and editing; MT, Conceptualization, Formal analysis; ES, Methodology, Contributed novel analytic tools and provided insights on their adaption for the present study; UH, Investigation, Methodology, Writing—review and editing, Contributed novel analytic tools and provided insights on their adaption for the present study; MB, Conceptualization, Resources, Supervision, Funding acquisition, Investigation, Methodology, Writing—original draft, Project administration, Writing—review and editing; GA, Conceptualization, Resources, Formal analysis, Supervision, Funding acquisition, Investigation, Methodology, Writing—original draft, Project administration, Writing—review and editing

## Author ORCIDs
Gideon Rosenthal, http://orcid.org/0000-0002-9783-0590
Marlene Behrmann, http://orcid.org/0000-0002-3814-1015
Galia Avidan, http://orcid.org/0000-0003-2293-3859

## Ethics
Human subjects: The experiment was approved by the Institutional Review Boards of Carnegie Mellon University (IRBSTUDY2015_00000191), and all participants provided signed informed consent. The experiment was approved by the Helsinki committee of the Soroka University Medical Center, Beer Sheva, Israel (5106) and all participants provided signed informed consent.

# Additional files

## Major datasets
The following dataset was generated:

| Author(s) | Year | Dataset title | Dataset URL | Database, license, and accessibility information |
|---|---|---|---|---|
| Gideon Rosenthal, Michal Tanzer, Erez Simony, Uri Hasson, Marlene Behrmann, Galia Avidan | 2017 | Altered topology of neural circuits in congenital prosopagnosia | https://doi.org/10.1184/R1/5774289.v1 | Available at Figshare under the Creative Commons Attribution 4.0 International (CC BY 4.0) license |

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
