## [Decision Letter]

Thank you for submitting your article "Altered topology of neural circuits in congenital prosopagnosia" for consideration by *eLife*. Your article has been reviewed by three peer reviewers, and the evaluation has been overseen by a Reviewing Editor and David Van Essen as the Senior Editor.

The reviewers have discussed the reviews with one another and the Reviewing Editor has drafted this decision to help you prepare a revised submission.

Summary:

The paper presents a new network analysis method, to characterize and compare the large scale network topology of brain networks in congenital prosopagnosia (CP) compared to typical controls. By using a new approach (inter-subject functional correlation approach, ISFC), which captures the correlations between nodes of different subjects, the study reports both local and macro scale differences between CP and controls. Locally, there were differences across groups in the anterior temporal (ATL), which served a hub for the control participants, but not CPs. Macroscopically, CPs evinced hyper-connectivity in posterior visual regions compared to controls, especially in the right hemisphere. Moreover, this hyper-connectivity was higher in the right hemisphere, known for its dominance in face processing, compared to the left hemisphere. These results advance understanding of network differences between CPs and controls.

The reviewers agreed that the topic is of great interest, the rationale of the study is straightforward, and the experiments were well designed. They concluded that the study offers new insights into the perturbed cortical topology in CPs.

Essential revisions:

The reviewers raised several major concerns that require your attention before the paper can be considered for publication.

Conceptual

The reviewers raised concerns about overlap with your prior 2014 Cerebral Cortex paper. Please clarify the conceptual advancement made by this study with respect to your prior work.

Another concern raised was that the comparison between normal controls and subgroups of CPs (mild and severe group) is non-independent from the comparison between normal controls and the entire group of CPs. Moreover, the results comparing the entire group of CPs and the different severity levels seems to be at odd with each other. The reviewer commented: "The authors report that compared with controls, CPs showed decreased ISFC in the FFA tagged nodes, as well as increased ISFC in the LOC tagged nodes. However, in the comparison between severe CPs and mild CPs, they report compared with mild CPs, severe CPs showed decreased ISFC in the LOC tagged nodes, as well as increased ISFC in the FFA tagged nodes. The explanations in the discussion pertaining to the inconsistency between these findings is speculative."

Data analysis

Thresholding procedure: The reviewers raised concerns regarding the threshold used, and asked why only positive correlations were considered:

1) Throughout the paper different thresholds are mentioned, but no clear description or quantification of how these thresholds were computed is provided. For example, Figure 1. uses an edge threshold of 0.3 without motivating why such a threshold is used, while the legend of Figure 2 reads on the second line "at minimum threshold", but it's unclear what the minimum is here.

2) Definition of nodes:

Why was a 0.5 threshold used? A better motivation should be provided.

Permutation testing: The permutation test that was used to evaluate the statistical differences of the network structure across the two groups was not properly implemented.

In permutation testing, given a statistic t computed on the difference between two groups, one would randomly assign the group labels, and compute the statistic again, repeat this process multiple times to estimate a null distribution, and compute the empirical p-value of the original statistic. If we apply the same concept to the method developed by the authors, one should first estimate the statistic, that is a correlation matrix for each group. This would be done by performing the split-half procedure for all the possible combinations (10C5 or 252 splits), and averaging across all splits, similar to what Simony et al. did with the leave-one-subject-out scheme. Given these two correlation matrices (one for each group), one would take their difference and obtain the final statistic of interest. Then, to compute the null distribution, one would randomly assign subjects to the groups (while maintaining the same numbers of subjects in each group), and then perform the same procedure to estimate the final statistic. By repeating this process at least 10,000 times (modern computing power can easily allow such number of iterations) one would effectively create a null distribution, that is the sampling distribution under the null hypothesis of no difference between the two groups. FWER correction could be performed in the same way as in Simony et al., 2016, by taking the max of the statistic at each iteration.

Estimation of the null distribution for statistical testing. In the original paper by Simony et al., ISFC was computed in a leave-one-subject-out scheme: time series from one subject were correlated with the average time-series of the other subjects. This process was repeated for all the possible combinations, and an average correlation matrix was computed, imposing symmetry by averaging the correlation matrix with its transpose. Statistical significance was computed through permutation testing, by randomizing the phase of the time series and computing an empirical, FWER corrected p-value. In contrast, Rosenthal et al., 2016 adopt what is known as a split-half scheme: for each group the correlation matrix is computed by correlating the average time series of five subjects with the average time series of the other five subjects. Statistical testing, however, is then mixed with the estimation of the actual correlation matrix from the data, by repeating this process 1,000 times and counting how many times the correlation in one group is greater than the correlation in the other group, to seemingly estimate an empirical p-value. The concern is that this approach might conflate the estimation of the correlation matrix with permutation testing, and might not compute a null distribution.

Comparison between functional connectivity (FC) analysis and ISFC: Comparing ISFC with FC to show the effectiveness of the novel method is very important, However, this comparison should be done while keeping all the statistical analyses comparable, similar to the approach of Simony et al., 2016.

In the present manuscript, the different methods used different statistical tests: t-test to compute significance in FC and a permutation testing in ISFC to evaluate differences between CPs and controls. It is unclear whether the usage of different statistical tests could lead to the differences across the two methods. A permutation testing approach in the analysis of FC could resolve this issue.

Additionally, for ISFC the time series of two runs were averaged and then the correlation matrix computed, whereas for FC correlation matrices were computed within each run, and then averaged. Consistency should be maintained (either averaging time series or computing correlation matrix individually for each run).

Brain-behavioral correlations: It would be interesting to link the topological changes to the severity of CP by conducting a correlation analysis between behavioral performance with topological measures, in which each point in the correlation is an individual subject. This approach could evaluate if the topological changes could predict behavioral performance in individual CPs. A similar analysis could also be conducted in normal groups.

[Editors' note: further revisions were requested prior to acceptance, as described below.]

Thank you for resubmitting your work entitled "Altered topology of neural circuits in congenital prosopagnosia" for further consideration at *eLife*. Your revised article has been favorably evaluated by David Van Essen (Senior editor), a Reviewing editor, and three reviewers.

The manuscript has been improved but there are some remaining issues listed by reviewers 2 and 3 that need to be addressed before acceptance, as outlined below. It is important to address these concerns incisively in the revised manuscript and in a point-by-point response in your cover letter.

Reviewer #1:

The authors have addressed my previous suggestions.

Reviewer #2:

I'm glad to see my comments have been appropriately addressed, and the quality of the paper has been significantly improved. Importantly, the new correlation analysis between the ISFC edges and behavioral performance really provide us new insights on the neural mechanism of the CP. But on second thought I still have several concerns that the authors may want to address.

1) I do agree that ISFC is a powerful approach to detect the stimulus-locked inter-regional correlation patterns. But as the visual experiment in the current study is a blocked design rather than the continuous natural (or real life) stimulus design as that in Simony et al., 2016, I'm wondering if the ISFC were mainly driven by the 'box car' effect rather than stimulus-locked response. If so, maybe the authors need to remove the data from the rest periods before running ISFC.

2) The data from an independent localizer was used to define the nodes. It elegantly avoids the double dipping. But I'm still wondering if there are some biases because only the data from the controls were taken into account in defining the nodes. Will this choice bias some results (e.g., the face-specific dominance in controls' nodes)?

3) In the definition of edges using the standard FC, why the author normalized all correlation coefficients for each subject to remove the subject level global effects? Aren't the global effects our interests in comparing controls and CPs? And, why the authors didn't use the same strategy to normalize all correlation coefficients in the definition of edges using the ISFC?

4) As I pointed out in the last round of review that the top schematic in Figure 1 and Figure 4 is somewhat misleading. The dash lines which connect the corresponding nodes in a pair of brains may mislead the readers into thinking that ISFC only calculates the functional correlation between corresponding nods across brain as inter-subject correlation (ISC) does. Nor do I catch what the top schematic in Figure 2 exactly does express.

Reviewer #3:

The authors addressed most of the comments in the first round of the review, however I believe that the manuscript can be improved further in clarity and in methodological rigor.

Definition of node strength. The authors abandoned the use of node degree in favor of weighted node degree, or node strength, by creating an undirected weighted graph, where the weights are t-values of correlations between pairs of ROIs. However, they used both negative and positive t-values at the same time to compute the strength of each node. As reported in the Appendix of the paper the authors cite (Rubinov and Sporns, 2010), the assumption of this metric is that all the weights are positive and normalized between 0 and 1. In this manuscript, weights are not all positive, and not normalized (although I believe this is less problematic). I'm not sure that it makes sense to consider both negative and positive weights at the same time. Imagine a node with only two incident edges, one with positive weight (say +10), and one with negative weight (-10). The strength of this node will be 0, but this doesn't correctly represent that the node is strongly correlated with another node only in the Control > CP contrast, and strongly correlated with a different node only in CP > Control. It is my understanding that in networks with negative weights one should analyze both sets of weights (positive and negative) separately.

Correlation between ISFC and behavior. I find that this analysis has several issues that need to be addressed. First, the authors used a one-tailed t-test, but this is generally considered too liberal without a clear a priori hypothesis on the directionality of the effect. It is even more problematic considering that with a two-tailed t-test, none of the results would be significant. In addition, even with a one-tailed t-test, the correlation between CFMT and ISFC values is not significant (even though the authors claim that it is: p = 0.053). Second, I find even more problematic the fact that the authors computed two different correlations and regressions, one for each group; looking at Figure 2 it is evident that if the two groups were considered together, the regression line would be flat or perhaps negative. This should be addressed by running a single multiple regression with a regressor variable indicating the group. Also, using z-scores as the x-axis in Figure 2 is confusing: there's no need to z-score the values prior to correlating them, and showing the actual questionnaire score will allow the reader to understand better how the participants score in those questionnaires. Lastly, what do the authors mean with "raw ISFC value"? Given the node by node ISFC matrix, does it refer to the average across one row of this matrix (perhaps taking only the upper triangular matrix if the ISFC matrix is symmetric)? And is the diagonal considered or removed prior to averaging, if the average is being computed?

Comparison between FC and ISFC. The authors improved the permutation testing approach for the ISFC, however they used an FDR corrected t-test with FC. The authors should consider implementing the permutation approach used in Simony et al., 2016, to make the two methods really comparable (i.e., differing only in the way correlations are computed), or justify why they chose these different statistical tests. In addition, ISFC correlations are still computed by averaging the time-series in the two runs first, whereas in FC first correlation matrices are computed, and then averaged. This should be made consistent in both analyses to make the comparison clearer.

In addition, it seems now that results from FC and ISFC converge. Then, why is ISFC needed at all? The authors should expand their discussion on this point. Also, correlations between edges and behavioral results should be performed also using FC-if not in the main text, they should be inserted in the supplementary materials to let readers understand the benefits/drawbacks of either methods when investigating similar questions.

Comparison between node strength and node location. Please provide more information on the stability of this result using different number of bins/bin sizes. How many bins had only one node? Also, for each bin, was the node strength computed as the sum of all the nodes belonging to that bin, or the average/median? The values in the y-axis in Figure 3 seem too high to be averages, and this analysis suffers from the problem described above of putting together negative and positive weights: a better approach would be to divide negative and positive edges, and perform the same analysis.

Please provide units for both axes in Figure 3.

The legend of Figure 3 refers to controls as "matched", but throughout the paper "controls" is used.

---

## [Author Response]

*Conceptual*

*The reviewers raised concerns about overlap with your prior 2014 Cerebral Cortex paper. Please clarify the conceptual advancement made by this study with respect to your prior work.*

We thank the reviewers for this important comment. Following the review process we feel that the distinction between the 2014 Cerebral Cortex had become clearer. Specifically, in contrast with the 2014 paper, in the current study, we utilized ISFC to investigate a large swath of visually related cortex in a relatively unbounded fashion. This led to the discovery of posterior hyper-connectivity in CP compared to control subjects. Furthermore, following the reviewers’ suggestions, we correlated the ISFC values with the measured behavioral measures and, we were pleased to discover that this analysis revealed a positive correlation with the two main behavioral markers of prosopagnosia. The anterior temporal hypo-connectivity together with the posterior hyperconnectivity and its correlation with behavior may explain individual differences between CP individuals.

*Another concern raised was that the comparison between normal controls and subgroups of CPs (mild and severe group) is non-independent from the comparison between normal controls and the entire group of CPs. Moreover, the results comparing the entire group of CPs and the different severity levels seems to be at odd with each other. The reviewer commented: "The authors report that compared with controls, CPs showed decreased ISFC in the FFA tagged nodes, as well as increased ISFC in the LOC tagged nodes. However, in the comparison between severe CPs and mild CPs, they report compared with mild CPs, severe CPs showed decreased ISFC in the LOC tagged nodes, as well as increased ISFC in the FFA tagged nodes. The explanations in the discussion pertaining to the inconsistency between these findings is speculative."*

The comparison of CP sub-groups employed in the previous version of the manuscript was used as a proxy for tapping into individual differences in CP. Following the reviewers’ helpful comments, we used a more powerful measure to compare directly the ISFC values with the individual behavioral measures. As noted above, we were able to discover a significant correlation between ISFC and behavioral performance. Hence, we think that the original division of CP to sub-groups had become redundant as well as too cumbersome, compared to the new analysis, and we have removed it from the current, new version.

*Data analysis*

*Thresholding procedure: The reviewers raised concerns regarding the threshold used, and asked why only positive correlations were considered:*

*1) Throughout the paper different thresholds are mentioned, but no clear description or quantification of how these thresholds were computed is provided. For example, Figure 1. uses an edge threshold of 0.3 without motivating why such a threshold is used, while the legend of Figure 2 reads on the second line "at minimum threshold", but it's unclear what the minimum is here.*

Following the reviewers’ comment, we conducted the analysis in two complementary phases: first we conducted an analysis in which we employed the stringent FDR correction for multiple comparisons. Note that this criterion practically dictates that in any comparison (Controls>CPs and vice versa) there would be no edges which exhibits the opposite pattern of ISFC. Next, in our second analysis, we did not use any thresholding and examined the ISFC pattern by employing weighted graph measures, thereby, avoiding arbitrary thresholding decisions. Note that this is a common practice among researchers in the field of network analysis.

*2) Definition of nodes:*

*Why was a 0.5 threshold used? A better motivation should be provided.*

Our main motivation for choosing the 0.5 threshold was that it allowed wide coverage of the cortex. We do not ascribe a unique meaning for this particular threshold, but rather applied it as a starting point that would allow a more widespread investigation. We refer to this issue in the paper in subsection “Definition of nodes”.

*Permutation testing: The permutation test that was used to evaluate the statistical differences of the network structure across the two groups was not properly implemented.*

*In permutation testing, given a statistic t computed on the difference between two groups, one would randomly assign the group labels, and compute the statistic again, repeat this process multiple times to estimate a null distribution, and compute the empirical p-value of the original statistic. If we apply the same concept to the method developed by the authors, one should first estimate the statistic, that is a correlation matrix for each group. This would be done by performing the split-half procedure for all the possible combinations (10C5 or 252 splits), and averaging across all splits, similar to what Simony et al. did with the leave-one-subject-out scheme. Given these two correlation matrices (one for each group), one would take their difference and obtain the final statistic of interest. Then, to compute the null distribution, one would randomly assign subjects to the groups (while maintaining the same numbers of subjects in each group), and then perform the same procedure to estimate the final statistic. By repeating this process at least 10,000 times (modern computing power can easily allow such number of iterations) one would effectively create a null distribution, that is the sampling distribution under the null hypothesis of no difference between the two groups. FWER correction could be performed in the same way as in Simony et al., 2016, by taking the max of the statistic at each iteration.*

*Estimation of the null distribution for statistical testing. In the original paper by Simony et al., ISFC was computed in a leave-one-subject-out scheme: time series from one subject were correlated with the average time-series of the other subjects. This process was repeated for all the possible combinations, and an average correlation matrix was computed, imposing symmetry by averaging the correlation matrix with its transpose. Statistical significance was computed through permutation testing, by randomizing the phase of the time series and computing an empirical, FWER corrected p-value. In contrast, Rosenthal et al., 2016 adopt what is known as a split-half scheme: for each group the correlation matrix is computed by correlating the average time series of five subjects with the average time series of the other five subjects. Statistical testing, however, is then mixed with the estimation of the actual correlation matrix from the data, by repeating this process 1,000 times and counting how many times the correlation in one group is greater than the correlation in the other group, to seemingly estimate an empirical p-value. The concern is that this approach might conflate the estimation of the correlation matrix with permutation testing, and might not compute a null distribution.*

Following the reviewers’ suggestions we applied a permutation test as suggested and have corrected for multiple comparisons using FDR (see subsection “ISFC Formulation”). We are pleased to report that the results remain unchanged. We thank the reviewers’ for raising this important issue.

*Comparison between functional connectivity (FC) analysis and ISFC: Comparing ISFC with FC to show the effectiveness of the novel method is very important, However, this comparison should be done while keeping all the statistical analyses comparable, similar to the approach of Simony et al., 2016.*

*In the present manuscript, the different methods used different statistical tests: t-test to compute significance in FC and a permutation testing in ISFC to evaluate differences between CPs and controls. It is unclear whether the usage of different statistical tests could lead to the differences across the two methods. A permutation testing approach in the analysis of FC could resolve this issue.*

*Additionally, for ISFC the time series of two runs were averaged and then the correlation matrix computed, whereas for FC correlation matrices were computed within each run, and then averaged. Consistency should be maintained (either averaging time series or computing correlation matrix individually for each run).*

Following the reviewers’ suggestions we applied the same statistical procedure and same averaging approach to the FC analysis. This procedure yielded similar overall results, while further supporting our main results showing a significant reduction of connectivity in one of the anterior temporal edges in the controls > CPs contrast. A more complete picture of this reduction is evident in the ISFC analysis.

*Brain-behavioral correlations: It would be interesting to link the topological changes to the severity of CP by conducting a correlation analysis between behavioral performance with topological measures, in which each point in the correlation is an individual subject. This approach could evaluate if the topological changes could predict behavioral performance in individual CPs. A similar analysis could also be conducted in normal groups.*

As noted above, this important comment led us to run a complete re-analysis of the data using ISFC with a one-vs-all comparison. Specifically, to be able to correlate behavioral in the individual level, each subject must have an idiosyncratic neuronal measure. Hence, we modified the split-half ISFC analysis used in the original manuscript, to the correlation of one subject with the mean of the remaining subjects (as detailed in Simony et al., 2016). This procedure uncovered a significant correlation between the ISFC obtained from the CPs > controls contrast and the behavioral performance on the diagnostic face task. We think that this finding further strengthen the paper and we thank the reviewers for their important contribution.

[Editors' note: further revisions were requested prior to acceptance, as described below.]

*Reviewer #2:*

*I'm glad to see my comments have been appropriately addressed, and the quality of the paper has been significantly improved. Importantly, the new correlation analysis between the ISFC edges and behavioral performance really provide us new insights on the neural mechanism of the CP. But on second thought I still have several concerns that the authors may want to address.*

*1) I do agree that ISFC is a powerful approach to detect the stimulus-locked inter-regional correlation patterns. But as the visual experiment in the current study is a blocked design rather than the continuous natural (or real life) stimulus design as that in Simony et al., 2016, I'm wondering if the ISFC were mainly driven by the 'box car' effect rather than stimulus-locked response. If so, maybe the authors need to remove the data from the rest periods before running ISFC.*

We thank you for this important comment. The reviewer is correct that the current study utilizes a block design and that it is different from the "real life" experiment performed in Simony et al., 2016. We certainly agree with the reviewer that examining the potential contribution of rest periods to the ISFC response is of interest. However, after careful consideration of this issue and the potential implications of the design employed in the present study, we came to the conclusion that it is unlikely that an on/off signal could account for the present findings. Moreover, we are of the view that the current design is not suitable for such an analyses. Below, we lay out our response to this comment and to the potential effects of a block-design on our findings:

· It is important to note that the block design paradigm is identical in the two experimental groups (CP, controls) and critically, the differences obtained between the groups were selective and were observed only in high level visual regions. It is unlikely that these selective differences would be driven by an on/off effect. If that were the case, we might expect to find group differences in early visual cortex, which would be mostly affected by an ‘on/off’ visual stimulations, rather than in the ATL and the LOC. Additionally, these particular results confirmed our hypothesis that the ATL would exhibit reduced ISFC in CP compared with controls which was based on previous structural and functional studies (Avidan et al., 2014; Odegard et al., 2009; Steinbrink et al., 2008; Thomas et al., 2009). Finally, the ISFC in the vicinity of the LOC was correlated with behavior. For these reasons, it is unlikely that the differences are an artifact or a product of an on/off blocked stimulation. Rather, the more likely interpretation is that the findings stem from a real difference in the stimulus-driven signal between the experimental groups.

· Note that Simony et al., 2016 replicated their continuous ISFC analysis with a control block experimental design to test whether ISFC can be implemented in non-continuous paradigms (Supplementary figure 5 in their paper). Although their block design analysis included concatenated stimulus blocks, the signals also suffered from possible on-off transitions, as the boundaries between the stimulus and resting blocks were not removed. That they were able to replicate the key result with the blocked paradigm confirms that the ISFC is capable of capturing the stimulus driven signal and not the transitions between the blocks.

· In relation to the previous point, we agree that even when dismissing these observations, whether the differences between the experimental groups are driven by the blocked stimuli (or in other words, the presentation of sets of stimuli) or by the individual trials is certainly of interest, but it does not contradict our main claim that the obtained ISFC is selective in a group dependent fashion. We certainly plan to specifically address this issue in our future studies.

· It is also important to note that when we first considered the analytical procedures for this study, we actually considered the approach suggested by the reviewer and concluded that although deconstruction of the ISFC may be of interest, the current experimental design does not permit this kind of analysis. The "resting" blocks in the current experiment are only 6 seconds long (3 TRs). The HRF response is sluggish and variable across regions and individuals. This signal mainly consists of two parts – a positive signal change that peaks at approximately 4 to 6 seconds post stimulus onset and a post stimulus undershoot that peaks 6 to 10 seconds after the stimulus ends, but can last up to 30 seconds or more (Buxton, Uludağ, Dubowitz and Liu, 2004; Cohen, 1997; Frahm, Krüger, Merboldt and Kleinschmidt, 1996; Krüger, Kleinschmidt and Frahm, 1996; Mayer et al., 2014). Hence, the "resting" blocks in the current paradigm are not clean from the stimulus driven signal, and it is not obvious which time points from the "resting" blocks should be removed (and in fact the stimulus blocks may be "contaminated" by the "rest" signal and vice-versa).

Given these considerations, we think that the analysis suggested by the reviewer is not suitable for the specific design we employed and would potentially induce confounds. Nevertheless, we do agree that a paradigm involving continuous stimulus presentation may help to deconstruct the ISFC into inter-trial vs block related signal, as we have previously reported in a different context (Rosenthal, Sporns and Avidan, 2016). If the reviewer feels strongly about this point, we can certainly re-do the analysis with the "resting blocks" removed.

*2) The data from an independent localizer was used to define the nodes. It elegantly avoids the double dipping. But I'm still wondering if there are some biases because only the data from the controls were taken into account in defining the nodes. Will this choice bias some results (e.g., the face-specific dominance in controls' nodes)?*

The reviewer is correct that the definition of the nodes is independent from the current experiment and that the nodes are extracted from control participants. Nevertheless, in previous studies, we showed that the location, BOLD activation and overall spread of activation of the face patches in core face regions of the face network in CP, is not different from controls (Avidan et al., 2013; Avidan et al., 2005; Avidan and Behrmann, 2009; DeGutis, Bentin, Robertson and D’Esposito, 2007; Hasson et al., 2003). Hence, a difference in the definition of the ROIs is unlikely to drive the effects, and there is a benefit of using an independent node definition to avoid double dipping. Additionally, since this definition was done based on control data, we were able to use a larger group of participants to increase the reliability of the node definition. It was not feasible to recruit a comparable independent group of CPs on which only to define ROIs.

*3) In the definition of edges using the standard FC, why the author normalized all correlation coefficients for each subject to remove the subject level global effects? Aren't the global effects our interests in comparing controls and CPs? And, why the authors didn't use the same strategy to normalize all correlation coefficients in the definition of edges using the ISFC?*

The focus of the current study was the application of the ISFC approach. In this approach, only signal that is synchronized across subjects is kept and this analysis revealed selective group dependent effects and did not reveal global effects (see SNR section). These findings thus imply that such global effects in the data are idiosyncratic and are not part of the critical, paradigm related signal. In contrast, subject level FC global effects may be induced by noise and intrinsic connectivity, which might mask the results and prevent a direct comparison with the ISFC. We therefore normalized the matrices to obtain the relative differences between the regions per subject and subsequently obtained the selective differences between the groups. This procedure actually allowed for a more direct comparison between the FC and ISFC analyses.

*4) As I pointed out in the last round of review that the top schematic in Figure 1 and Figure 4 is somewhat misleading. The dash lines which connect the corresponding nodes in a pair of brains may mislead the readers into thinking that ISFC only calculates the functional correlation between corresponding nods across brain as inter-subject correlation (ISC) does. Nor do I catch what the top schematic in Figure 2 exactly does express.*

We thank the reviewer for this clarification. We have now modified the schematics as per the reviewer's request and we hope these are clearer now.

*Reviewer #3:*

*The authors addressed most of the comments in the first round of the review, however I believe that the manuscript can be improved further in clarity and in methodological rigor.*

*Definition of node strength. The authors abandoned the use of node degree in favor of weighted node degree, or node strength, by creating an undirected weighted graph, where the weights are t-values of correlations between pairs of ROIs. However, they used both negative and positive t-values at the same time to compute the strength of each node. As reported in the Appendix of the paper the authors cite (Rubinov and Sporns, 2010), the assumption of this metric is that all the weights are positive and normalized between 0 and 1. In this manuscript, weights are not all positive, and not normalized (although I believe this is less problematic). I'm not sure that it makes sense to consider both negative and positive weights at the same time. Imagine a node with only two incident edges, one with positive weight (say +10), and one with negative weight (-10). The strength of this node will be 0, but this doesn't correctly represent that the node is strongly correlated with another node only in the Control > CP contrast, and strongly correlated with a different node only in CP > Control. It is my understanding that in networks with negative weights one should analyze both sets of weights (positive and negative) separately.*

As per the reviewer's request, we performed the analyses using positive and negative weights separately, which produced consistent findings with those reported in the previous version of the manuscript. Normalization of the weights is not essential and do not affect the results in our analyses as we use the relative strength in some analyses and the correlation with other measures in other analysis. The absolute strength values in this case is of no relevance (Barrat, Barthélemy, Pastor-Satorras, and Vespignani, 2004). See details in subsection “Network analysis” and Figure 2

*Comparison between node strength and node location. Please provide more information on the stability of this result using different number of bins/bin sizes. How many bins had only one node? Also, for each bin, was the node strength computed as the sum of all the nodes belonging to that bin, or the average/median? The values in the y-axis in Figure 3 seem too high to be averages, and this analysis suffers from the problem described above of putting together negative and positive weights: a better approach would be to divide negative and positive edges, and perform the same analysis.*

The original analysis used the mean of the nodes. We now split the analysis to negative and positive edges as suggested by the reviewer and, importantly, we report consistently similar results across various bin divisions from 10 to 21. We are very pleased to see the stability of the findings across the different approaches. See details in subsection “Network analysis” and Figure 2

*Correlation between ISFC and behavior. I find that this analysis has several issues that need to be addressed. First, the authors used a one-tailed t-test, but this is generally considered too liberal without a clear a priori hypothesis on the directionality of the effect. It is even more problematic considering that with a two-tailed t-test, none of the results would be significant. In addition, even with a one-tailed t-test, the correlation between CFMT and ISFC values is not significant (even though the authors claim that it is: p = 0.053). Second, I find even more problematic the fact that the authors computed two different correlations and regressions, one for each group; looking at Figure 2 it is evident that if the two groups were considered together, the regression line would be flat or perhaps negative. This should be addressed by running a single multiple regression with a regressor variable indicating the group.*

Following the reviewer's comment and suggestions, we re-examined our data and models and this led to several modifications of the analysis. First, based on previous studies showing correlation between CFMT and the famous faces questionnaire (Furl, Garrido, Dolan, Driver and Duchaine, 2011) and some recent data from our own lab (Mardo, Hadad and Avidan, 2017 under review), we started off by examining whether the two behavioral measures are redundant and subsequently, whether there is a latent behavioral measure which can account for both. In accordance with these studies, we found a significant correlation between the CFMT and famous faces behavioral measures *r*(18)=0.71, *p*=0.0003. Accordingly, the 2 measures were factorized using PCA (Furl et al., 2011), and the first principal component which accounted for 85.9% of the variance served as our new face recognition behavioral score. The correlation between the face recognition behavioral score was *r*(18) = 0.9, *p<0.0001* and *r*(18) = 0.94, *p<0.0001* for CFMT and famous faces questionnaire respectively.

Following the reviewer's suggestion a multivariate regression was conducted to examine if ISFC and group predicted the face recognition behavioral score. This was done for the significant ISFC edges obtained from both the CP>Controls and the Controls>CP contrasts. As for the CP>controls contrast, overall a significant effect was found for both the group and the ISFC edge and the independent variables indeed explained a significant amount of the face recognition behavioral score *(R^2^ =0.83, R^2^_Adjusted_ =.81, F(2,17) = 42.67, p < 0.0001).* Moreover, the group significantly predicted face recognition behavioral score (β = -6.08, *p* < 0.0001), as did the ISFC value (β = 4.89, *p* =0.017). Hence, ISFC score was positively associated with behavioral face performance. When we conducted the same procedure but examining the edges obtained from the controls>CPs contrast, the ISFC edge and the independent variables also explained a significant amount of the variance of the face recognition behavioral score *(R^2^ =0.77, R^2^_Adjusted_ = 0.74, F(2,17) = 29.19, p < 0.0001).* Nevertheless, only the group effect coefficient was significant (β = -5.23, p = 0.005), while the ISFC coefficient was not (β = -5.12, p = 0.44). Thus, indicating the lack of linear relation between the behavioral measure and ISFC in this contrast.

This analysis was added in subsection “Correlations of the significant edges with behavior” and we thank the reviewer for this comment. This has strengthened our conclusions.

*Also, using z-scores as the x-axis in Figure 2 is confusing: there's no need to z-score the values prior to correlating them, and showing the actual questionnaire score will allow the reader to understand better how the participants score in those questionnaires.*

As we chose to use an aggregated measure for behavioral face recognition performance, this comment is no longer relevant. Accordingly, we have removed the correlation figure from the manuscript.

*Lastly, what do the authors mean with "raw ISFC value"? Given the node by node ISFC matrix, does it refer to the average across one row of this matrix (perhaps taking only the upper triangular matrix if the ISFC matrix is symmetric)? And is the diagonal considered or removed prior to averaging, if the average is being computed?*

The raw ISFC is the value of the significant edges in the CP>control or control>CP contrasts which is indeed the average of the lower and upper triangle edges of the ISFC matrix. This is not the average of a row, but a single edge and hence, the diagonal is of no relevance.

*Comparison between FC and ISFC. The authors improved the permutation testing approach for the ISFC, however they used an FDR corrected t-test with FC. The authors should consider implementing the permutation approach used in Simony et al., 2016, to make the two methods really comparable (i.e., differing only in the way correlations are computed), or justify why they chose these different statistical tests.*

In the current study, we had a specific hypothesis regarding the anterior temporal cortex hypo-connectivity and did not want to rely on a family-wise correction. The FDR procedure allowed us to test the edge-wise statistically significant difference between the groups.

*In addition, ISFC correlations are still computed by averaging the time-series in the two runs first, whereas in FC first correlation matrices are computed, and then averaged. This should be made consistent in both analyses to make the comparison clearer.*

We thank the reviewer for the comment. The time series in the two runs were first averaged and only then the FC correlation matrices were computed. This modification was made in the previous revision but was not properly described in the Materials and methods section. We now corrected this.

*In addition, it seems now that results from FC and ISFC converge. Then, why is ISFC needed at all? The authors should expand their discussion on this point. Also, correlations between edges and behavioral results should be performed also using FC-if not in the main text, they should be inserted in the supplementary materials to let readers understand the benefits/drawbacks of either methods when investigating similar questions.*

We repeated the multiple regression analysis as described above. Specifically, we used the edges obtained from the FC as opposed to the ISFC and group independent variables to predict the face recognition behavioral score. This was done for the significant FC edges obtained from both the CP>Controls and the Controls>CP contrasts. The regression model with FC and the group independent variables explained a significant amount of the variance of the face recognition behavioral score for both CPs>controls *(R^2^ =0.78, R^2^_Adjusted_ = 0.75, F(2,17) = 30.17, p < 0.0001)* and controls>CPs *(R^2^ =0.78, R^2^_Adjusted_ = 0.75, F(2,17) =30.78, p < 0.0001)*. Nevertheless, for both the CPs>Controls and Controls>CPs only the group effect coefficient was significant (β = -4.99, *p* < 0.0001) and (β = -5.23, *p* < 0.0001) respectively, while the FC coefficient was not (β =0.95, *p* = 0.31) and (β =1.34, *p* = 0.25) respectively. Thus, no direct link was found between the face recognition behavioral score and FC connectivity. A previous study has shown that the ISFC better captures momentary network configurations and better predicted behavioral performance (Simony et al., 2016).

Consistently with this study, in the context of the present study, only the ISFC, but not FC was shown to be related to the behavioral measure of face processing, and hence was able to elucidate an additional important aspect of the data

The analysis was added in subsection “Correlations of the significant edges with behavior”.

*Please provide units for both axes in Figure 3.*

*The legend of Figure 3 refers to controls as "matched", but throughout the paper "controls" is used.*

Thank you for the comments, the figure was corrected accordingly.

References:

Barrat, A., Barthélemy, M., Pastor-Satorras, R., & Vespignani, A. (2004). The architecture of complex weighted networks. *Proceedings of the National Academy of Sciences of the United States of America, 101*(11), 3747–3752. https://doi.org/10.1073/pnas.0400087101

Buxton, R. B., Uludağ, K., Dubowitz, D. J., & Liu, T. T. (2004). Modeling the hemodynamic response to brain activation. *NeuroImage, 23, Supplement 1*, S220–S233. https://doi.org/10.1016/j.neuroimage.2004.07.013

Cohen, M. S. (1997). Parametric Analysis of fMRI Data Using Linear Systems Methods. *NeuroImage, 6*(2), 93–103. https://doi.org/10.1006/nimg.1997.0278

DeGutis, J. M., Bentin, S., Robertson, L. C., & D’Esposito, M. (2007). Functional plasticity in ventral temporal cortex following cognitive rehabilitation of a congenital prosopagnosic. *Journal of Cognitive Neuroscience*, *19*(11), 1790–1802. https://doi.org/10.1162/jocn.2007.19.11.1790

Frahm, J., Krüger, G., Merboldt, K. D., & Kleinschmidt, A. (1996). Dynamic uncoupling and recoupling of perfusion and oxidative metabolism during focal brain activation in man. *Magnetic Resonance in Medicine, 35*(2), 143–148.

Krüger, G., Kleinschmidt, A., & Frahm, J. (1996). Dynamic MRI sensitized to cerebral blood oxygenation and flow during sustained activation of human visual cortex. *Magnetic Resonance in Medicine, 35*(6), 797–800. https://doi.org/10.1002/mrm.1910350602

Mayer, A. R., Toulouse, T., Klimaj, S., Ling, J. M., Pena, A., & Bellgowan, P. S. F. (2014). Investigating the Properties of the Hemodynamic Response Function after Mild Traumatic Brain Injury. *Journal of Neurotrauma, 31*(2), 189–197. https://doi.org/10.1089/neu.2013.3069